# Follow the Mutations: Toward Class-Specific, Small-Molecule Reactivation of p53

**DOI:** 10.3390/biom10020303

**Published:** 2020-02-14

**Authors:** Stewart N. Loh

**Affiliations:** Department of Biochemistry and Molecular Biology, State University of New York Upstate Medical University, Syracuse, NY 13210, USA; lohs@upstate.edu; Tel.: +1-315-464-8731

**Keywords:** folding, stability, structure, zinc binding, DNA binding, metallochaperone, aggregation, tumor suppressor, cancer

## Abstract

The mutational landscape of p53 in cancer is unusual among tumor suppressors because most of the alterations are of the missense type and localize to a single domain: the ~220 amino acid DNA-binding domain. Nearly all of these mutations produce the common effect of reducing p53’s ability to interact with DNA and activate transcription. Despite this seemingly simple phenotype, no mutant p53-targeted drugs are available to treat cancer patients. One of the main reasons for this is that the mutations exert their effects via multiple mechanisms—loss of DNA contacts, reduction in zinc-binding affinity, and lowering of thermodynamic stability—each of which involves a distinct type of physical impairment. This review discusses how this knowledge is informing current efforts to develop small molecules that repair these defects and restore function to mutant p53. Categorizing the spectrum of p53 mutations into discrete classes based on their inactivation mechanisms is the initial step toward personalized cancer therapy based on p53 allele status.

## 1. Introduction

Mutation of the gene encoding for p53—a transcription factor and one of the cell’s master tumor suppressors—is the most recurrent genetic alteration in human cancer [1,2]. Nearly all these mutations cause p53 to lose its ability to properly recognize DNA and/or activate expression of its large complement of other tumor-suppressing genes [3,4]. In addition, a subset of p53 mutations appears to also act by introducing an oncogenic gain-of-function (GOF) by as-yet undefined pathways [5]. Consequently, a longstanding priority in cancer therapy is to develop drugs that restore transcriptional activity to mutant p53s, as well as inhibit their cancer-driving GOF properties. Because p53 is inactivated in many cancers, such drugs would likely have broad clinical impact. Indeed, it has been demonstrated that restoring proper function to mutant p53 is sufficient to kill a variety of tumors in mouse models [6,7,8].

The mutational landscape of p53 has several singular features that would seem to lend themselves to achieving the above goal. The majority of p53 tumorigenic mutations are of the missense variety (http://p53.iarc.fr), as opposed to the nonsense, frameshift, and deletion types that predominate in other tumor suppressors such as BRCA1/2 (http://arup.utah.edu/database/BRCA), retinoblastoma-1 (http://rb1-lovd.d-lohmann.de), and PTEN (http://www.lovd.nl/PTEN). Efforts to study p53 mutants are greatly aided by the fact that ~90 % of them map to a single domain, the ~220-amino acid DNA-binding domain (DBD), with one-third occurring at one of eight hot-spot codons in DBD [9]. To that end, the solution of the X-ray crystal structure of DBD in 1994 [10] and almost three decades of subsequent biophysical and biochemical analysis have defined some of the primary mechanisms by which the hot-spot mutations cause p53 to lose function (*vide infra*).

Why, then, are there no mutant p53-targeted drugs available to treat cancer patients? There are at least two root causes. First, there are few examples of drugs that target non-enzymatic, non-receptor proteins. In the case of p53, the drug would need to activate the mutant proteins but not stimulate wild-type (WT) p53, as the latter risks massive cell death due to widespread apoptosis and senescence [11]. This prospect is inherently more difficult than inhibiting protein function, the more common mode of action taken by the conventional drugs mentioned above. The second challenge, and that to which this review is directed, is that mutations act by different mechanisms, and therefore a drug that reactivates one mutant type may have no effect on another. There is a pressing need to classify mutations according to their effects on the structural, biophysical, and biochemical properties of p53, as well as on clinical features such as tumor development, metastasis, and response to therapy [12]. Only by doing so can the promise of treating cancer patients with drugs tailored to their p53 allele status start to become reality.

Here, we review the mechanisms by which missense mutations inactivate p53, strategies for categorizing mutants, and recent progress towards rescuing mutants by small molecules, interpreted through the lens of those classifications. We limit the focus to studies that target mutant DBD with the intended effect of restoring WT-like conformation and function. GOF remains poorly understood and is somewhat controversial [13], and, in general, a different set of tools is utilized when endeavoring to suppress GOF activities. These and other orthogonal approaches, such as selectively degrading mutant p53, disrupting interactions with p53 binding partners, inhibiting downstream signaling pathways, and exploiting synthetic lethality, have been reviewed recently [5,14,15]. We note, however, that restoring WT function to p53 mutants will serve to eliminate mutant GOF as well2. Three Classes of Tumorigenic p53 Mutations 

p53 consists of three domains: N-terminal transactivation (TAD; residues 1–89), central DNA binding (DBD; residues 94–312 (Figure 1)), and C-terminal oligomerization (OD; residues 95–393). Tetramerization and nuclear entry sequences are encoded in the OD. Once p53 has bound to DNA, the TAD is responsible for recruiting general transcription factors and RNA Pol II for gene expression [16]. TAD is also the primary site of MDM2 binding, the E3 ligase that works with MDMX to ubiquitinate p53 (at the OD) and flag it for proteasomal degradation. Compounds that disrupt the p53-MDM2/MDMX axis [17] are the subject of more clinical trials than any other class of p53-related drugs, although their application is mainly for cancers with WT p53 and they bind to MDM2 rather than p53.

DBD is the largest of the domains and home to ~90 % of known tumorigenic mutations. It is the only domain that does not contain extensive regions of intrinsic disorder, which has facilitated its high-resolution structural determination (Figure 1). Fersht and colleagues were the first to categorize DBD mutants based on their structural, biophysical, and biochemical properties [18,19,20,21]. The **DNA-contact class** is defined by a reduction in DBD’s affinity for DNA (1/*K_d_*^DNA^) without a reduction in its affinity for its single essential zinc ion (1/*K_d_*^Zn^) or its free energy of folding (ΔG_fold_) in the absence of zinc. A DNA contact mutation is suggested by its presence in the DNA-binding groove and is confirmed by measuring *K_d_*^DNA^ by electromobility shift or fluorescence assay. Alterations in any residue in direct contact with DNA have thus far been found to increase *K_d_*^DNA^. Like all transcription factors, p53 employs positively charged side chains to interact with the polyanionic DNA backbone, and mutations of Arg residues (such as R248Q and R273H; Figure 1) are a recurrent theme in human cancers (Table 1). In addition to being common, these mutants are the most pernicious: there is not yet an established paradigm for rationally designing molecules that restore proper DNA-binding affinity.

The **stability class** of mutants is characterized by the loss of folding free energy without a reduction in DNA- or Zn^2+^-binding affinity under the conditions in which the protein is folded. Destabilization is measured quantitatively by chemical denaturation of the zinc-free protein at low temperature (ΔG_fold_), or qualitatively by apparent melting temperature (T_m_; DBD aggregates at elevated temperatures and thermal denaturation is typically irreversible). Stability mutations are frequently found in the β-sandwich core of DBD, distant from both the DNA- and Zn^2+^-binding sites, with prime examples being Y220C (Figure 1; Table 1) and V143A, a well-known, temperature-sensitive mutant. DBD is only just folded at physiological temperatures (T_m_ 42–45 °C), and reducing ΔG_fold_ by as little as 1 - 2 kcal/mol is sufficient to cause appreciable unfolding, aggregation, and loss of function at physiological temperatures [19,25]. 

The **zinc-binding class** of mutations was originally inferred by their proximity to the metal-binding site [20]. The zinc ion is coordinated by C176, H179, C238, and C242; mutations of C176 and H179 are the 7th and 8th most common in the p53 database (Table 1). The removal of Zn^2+^ elicits structural changes, particularly in the loop that contains the DNA contact residue R248, that results in loss of DNA-binding specificity [22]. It also destabilizes DBD, causing it to cycle more rapidly between native and unfolded states [26]. The net result is a progressive loss of function even under native conditions, because a fraction of zinc-free DBD (apoDBD) molecules becomes kinetically trapped in misfolded, aggregation-prone conformations with each unfolding–folding cycle [25,26,27]. Although it is true that mutations in the metal-binding pocket will almost certainly weaken zinc affinity, it is possible that other mutations may also increase *K_d_*^Zn^ and be mistakenly assumed to belong to one of the other classes due to their distance from the zinc site. This underscores the need to classify mutants based on functional and biophysical data as well as on structural location, as discussed below.

## 2. Crosstalk Between Mutational Classes 

The three mutational classes involve distinct types of structural defects. These mutations, however, exert certain shared effects on folding and binding and it may be possible to exploit these commonalities in drug development. For example, the binding of zinc, like that of any ligand, is inextricably linked to protein folding and vice versa. Expressing this relationship as two coupled equilibria (Equation (1) and Equation (2)) helps to illustrate how zinc-binding and stability mutations are related. Zinc-binding mutants weaken metal binding (1/*K_d_*^Zn^ decreases; Equation (2)) without disrupting any interactions that affect the folding of the protein the in absence of metal (*K_fold_* remains constant; Equation (1)). By contrast, stability mutants introduce core packing or other defects that shift the population to the unfolded form (*K_fol_*_d_ decreases) but do not affect the ability of the folded protein to bind zinc (*K_d_*^Zn^ remains the same). *K_d_*^Zn^ and *K_fold_* can be measured independently [23] and, by doing so, zinc-binding mutants can be identified and their severity quantified, regardless of their location in the structure. Both mutational classes produce the same net effect of decreasing the population of functional DBD (DBD_folded_). The interdependence of Equation (1) and Equation (2) suggests that drugs that elevate intracellular zinc concentration ([Zn^2+^]_free_) may also refold stability mutants, and drugs that stabilize DBD may also rescue zinc-binding mutants, or reactivate p53 that has been demetallated by abnormally low [Zn^2+^]_free_.
(1)$$apoDBD_{unfolded}⇄apoDBD_{folded};\;K_{fold}$$
(2)$$apoDBD_{folded}+Zn^{2+}⇄DBD_{folded};\;1/K_{d}^{Zn}$$

The mutational categories are not mutually exclusive. Due to the cooperativity of the protein structure, a given alteration often affects more than one property, and this knowledge may help direct potential treatments. For instance, R249 is not in contact with DNA, and its substitution by Ser reduces ΔG_fold_ by 1.9 kcal/mol. The structural effects of R249S propagate to the DNA binding site [20], however, making it a mixed-stability/DNA-contact mutation that would benefit from eventual combination therapy. 

Mutation-induced aggregation/amyloid formation is a class-crossing pathway by which p53 loses function. p53 aggregation has long plagued in vitro experiments [19,22] but only recently have its underlying mechanisms been characterized and its relevance been extended to human tumor tissue. Generally, a soluble protein aggregates as a result of partial unfolding and subsequent exposure of a cryptic ‘sticky’ segment. One such amyloidogenic sequence was identified as residues 252–258 in DBD, which form a beta strand in the native structure [28,29]. As mentioned, zinc-binding and stability class mutations increase both the equilibrium population of partially unfolded forms and the rate at which they accumulate during the natural unfolding–folding cycle. Purified WT DBD forms amyloid fibrils under moderately destabilizing conditions [30] and aggregation is exacerbated by the hot-spot mutations Y220C [31], R175H [32], and R248Q [33]. An ex vivo study of biopsied human tumors found evidence of p53 aggregation (observed as punctate structures accumulating in the nucleus) that correlated with the presence of DBD mutations, including R175H and H179Y (zinc-binding class), Y220C (stability class), and R248L and R273H (DNA-binding class) [34]. Parenthetically, the authors noted that p53 aggregation induced a heat-shock-like stress response and speculated that the resulting dysregulation of protein homeostasis may contribute to GOF phenotypes and the apparent addiction of tumor cells to mutant p53. Even WT p53 was found to form nuclear inclusions under conditions where its transcriptional activity was compromised, suggesting that an elevated intracellular p53 protein load (caused by diminished MDM2 expression,) coupled with its inherent instability, are sufficient to drive aggregation. 

## 3. Rescuing Stability Class Mutants: Cavity Binders 

The so-called cavity binders are designed to selectively interact with the native form of the target protein and not with unfolded or misfolded forms (or with other proteins). Analogous to zinc binding (Equation (2)), drug–protein binding energy is used to drive the refolding of an unstable mutant. A drug that binds with a low *K*_d_ value is doubly important here, not only for the usual reason of minimizing dosage but also because higher affinity confers the ability to refold more severely destabilized mutants at any given drug concentration. The development of cavity binders has been challenged by the general lack of unique crevices or pockets on the surface of DBD, other than the DNA-binding groove itself. One notable exception is the small cavity left by truncation of the Tyr side chain in the Y220C stability class mutant (Figure 1). The carbazole derivative PK083 (Figure 2) was the first molecule developed to capitalize on this mutant-specific lesion, binding to Y220C p53 with *K_d_* ~150 μM, increasing its T_m_, slowing its rates of unfolding and aggregation, and partially restoring its apoptotic activity in cultured cells [35,36]. 

The affinity of Y220C cavity binders has progressively improved in recent years. Halogenating the pharmacophore scaffolds enhances their interaction with Lewis bases (such as the thiolate of C220) via halogen bonding [37]. Trifluorinating the N-ethyl group in the carbazole ring of PK083 lowered its *K_d_* by 5-fold [38]. Screening a library of iodinated aromatic compounds identified a new iodinated phenol chemotype that bound Y220C with *K_d_* = 184 μM, and chemical modification reduced this value to 9.7 μM (PK5196) [39]. Very recently, optimization of the carbazole and iodophenol cores yielded PK9318/PK9328 [40] and MB710 [41] (Figure 2), respectively, which binds Y220C with affinities in the 2–4 μM range and increases T_m_ by 2–4 °C. These compounds achieved >50 % killing of several p53–220C cancer cell lines at concentrations of 10–30 μM. MB710 was reasonably well-tolerated by WT p53 cell lines, but PK9318 and PK9328 demonstrated significant toxicity toward p53 knockout control cells, illustrating the need to further optimize the affinity and selectivity of binding.

Despite the steady gains made toward developing Y220C-specific drugs, a major gap remains in the ability to target other stability-class mutants for which no obvious pocket exists in static X-ray crystal structures. Computational simulations are revealing the dynamic and heterogeneous nature of conformational space that proteins naturally explore, especially when they are on the verge of unfolding, as is the case with DBD at 37 °C. These motions expose transient sites that can potentially be drugged. Combined molecular dynamics (MD)/experimental studies of R175H [42] and Y220C [43,44] DBD found evidence for previously hidden subpockets, the latter of which were useful in the aforementioned affinity optimizations. More recently, MD simulations of WT DBD and the stability-class V143A mutant found differences in the turn region between two beta strands (residues 208–213) [45]. The turn adopts an ‘open’ conformation in V143A that the authors hypothesized could be exploited for the rescue of this mutant, as well as other members of the stability class. On the experimental side, phage display-panning of random peptide libraries yielded 7–12 amino acid peptides (pCAPs) that appear to preferentially bind to the native state of p53 (WT, R175H, R249S, and V143A variants) and stabilize it against unfolding [46]. One of the peptides was shown to bind WT DBD with *K_d_* ~ 21 μM, although the site of interaction has not yet been determined.

## 4. Rescuing Zinc-Binding Class Mutants: Zinc Metallochaperones 

One can envision two basic mechanisms by which drugs can reinstate proper metal binding status to zinc-binding mutants. The first is binding to a secondary location on p53, changing the conformation of the zinc binding site, and restoring *K_d_^Zn^* to the wild-type value. The other is raising intracellular [Zn^2+^]_free_ to concentrations greater than *K_d_^Zn^* of the mutant p53. No molecule possessing the former activity has yet been discovered, but several ‘zinc metallochaperones’ (ZMCs) have been identified that act by means of the latter. In addition to the high-affinity, native Zn^2+^-binding pocket, DBD contains weaker, non-native sites, presumably comprised of its other seven Cys and nine His residues. Zinc misligation to these non-native sites traps DBD in a misfolded state and causes it to aggregate [26]. A ZMC is defined as a molecule that: (i) delivers zinc into the cell (ionophore activity), and then (ii) maintains intracellular Zn^2+^_free_ at concentrations appropriate for remetallating and refolding mutant DBD (zinc buffering activity) [47]. Zinc buffering is established by adjusting the *K_d_^Zn^* of the ZMC higher than that of the native site on the given mutant and lower than the collective *K_d_^Zn^* of the non-native sites, which is assumed to be similar from mutant to mutant. 

Zinc metallochaperone-1 (ZMC1) was the first small molecule shown to reactivate mutant p53 by the above mechanism [48]. One ZMC1 provides a Zn^2+^ half-site via the sulfur, β-nitrogen, and pyridinyl nitrogen from the thiosemicarbazone scaffold (Figure 3). Upon interaction with zinc, ZMC1 deprotonates and enolizes, creating the active (ZMC1)_2_Zn complex (Figure 3), the charge-neutrality of which is thought to be important for its ionophore activity [49]. ZMC1 transiently raises intracellular [Zn^2+^_free_] to levels (10–20 nM) that are appropriate for refolding R175H p53 and several other members of the zinc-binding class [49]. In concordance, ZMC1 induces R175H-specific killing of tumor cell lines and human tumor xenografts [50,51]. Several closely related molecules (the thiosemicarbazone ZMC2 and the selenosemicarbazone ZMC3) possess similar activities [52]. The semithiocarbazone COTI-2 may exert its antitumor effects at least in part through a ZMC mechanism, although it also induces apoptosis by p53-independent pathways [53,54]

In addition to ionophore and zinc buffering, a third activity of ZMC1—reactive oxygen species (ROS) generation—seems to play a role in p53 activation and cell death. ROS stimulates damage response pathways that post-translationally modify p53 to enhance its nuclear entry, recruitment of transcriptional machinery, and resistance to turnover [55]. Zinc ions exist only in a single oxidation state and cannot generate ROS, but it was shown that ZMC1 binds copper ions in the cell and creates ROS by as-yet uncharacterized reactions [56]. In agreement, cellular ROS scavengers such as N-acetyl cysteine do not affect ZMC1’s power to refold mutant p53, but inhibit the ability of the newly-reconformed protein to carry out its transcriptional program [23]. A recent study suggested that this property of ZMCs may be exploited to localize their potency to tumor sites. An alternative zinc-binding scaffold (nitrilotriacetic acid) was developed that reduced copper affinity by ~10^6^-fold relative to ZMC1 [57]. The cancer-cell-killing effect of these molecules synergized with chemotherapeutic agents and radiation, which served to supply ROS for p53 post-translational modification. One clearly must be alert to off-target toxicity caused by ZMC-mediated ROS, although these effects are not always undesirable. The aforementioned ZMC1-copper study found that ZMC1 arrested patient-derived glioblastoma cells at picomolar concentrations, due to ROS-induced depletion of deoxyribosyl purines [56].

Miller et al. took a creative approach by combining ZMC and cavity-binding functionalities in a single molecule [58]. A Zn^2+^-chelating pyridylmethyl moiety was mated to the Y220C-binding iodophenol core to create compound L5 (Figure 3). Its zinc-binding properties were consistent with those of ZMC1 and the compound elevated intracellular [Zn^2+^]_free_ levels, as expected. Treating a p53-Y220C stomach cancer cell line with L5 increased expression of p53 target genes, suggesting partial restoration of p53 function. L5 was reasonably cytotoxic towards this cancer cell line (IC_50_ = 1.7 μM) but also killed a WT p53 cell line at a similar dosage, again stressing the need for greater selectivity toward mutant p53 and reducing off-target effects. It is likely that the poor affinity of the starting scaffold for Y220C (*K_d_* ~200 μM [39]) is at least partly to blame.

## 5. Rescuing DNA Contact Mutants 

DNA-binding mutations can either remove a few protein–DNA contact points without altering the overall structure of the DNA-binding groove (e.g., R273H and R248Q) or distort the DNA-binding groove without changing any of the residues that contact DNA in WT DBD (e.g., R249S) [59]. In either case, there is no established strategy for replacing lost protein–DNA interactions using exogenous molecules, and this class is considered the most challenging to drug. Nevertheless, intragenic suppressor mutations have been shown to rescue transcriptional functions of R273H and R249S p53 in mammalian cells [60], suggesting that the conformation of the DNA-binding pocket may be somewhat malleable and subject to change by interactions at a distant site [59]. Indeed, an NMR study of DBD found increased dynamic motions in R273H DBD, compared to R248Q and WT, that spanned the DNA-binding surface and the β-sandwich core [61]. The authors concluded that R273H exhibits properties of a both a DNA-contact and conformational mutation, whereas R248Q is a classic DNA-contact mutation.

Very recently, a tryptophanol-derived oxazoloisoindolinone (SLMP53-1; Figure 4) was reported to activate the transcriptional function of the DNA-contact mutant R280K, as well as other hot-spot variants of p53 [62]. Thermal shift assays showed that SLMP53-1 bound directly to R280K DBD with an apparent *K_d_* of 10.7 μM. Molecular docking simulations of the p53 tetramer-DNA complex predicted that SLMP53-1 forms a sandwich-like interaction between Met243s of DBD dimers (on the opposite side of the DNA from R280) and several DNA bases, potentially buttressing the DBD–DNA interaction and partially compensating for the loss of hydrogen bonds from R280 to a guanine base in the major groove. In cell culture assays, SLMP53-1 rescued other p53 mutants from the DNA-contact class (R248Q, R248W, R273H, R282Q), zinc-binding class (R175H), and stability class (G245S). This result is not inconsistent with the proposed mode of action, because strengthening the DBD–DNA interaction may be expected to enhance the transcriptional function of p53 regardless of mutation. In fact, SLMP53-1 also activates WT p53 for cell killing [63], an effect that would ideally need to be mitigated in future iterations of this compound.

## 6. Rescuing p53 Aggregation 

Aggregation can be considered as a sink that draws p53 mutants into a nonfunctional state that’s not only refractory to rescue, but conceivably oncogenic due to GOF. All three p53 mutational classes are associated with aggregation, via their common effect of elevating intracellular p53 concentrations as well as through enhancing populations of aggregation-prone forms (zinc-binding and stability mutants). Molecules that inhibit aggregation therefore have the potential to treat a broad spectrum of p53 mutants [64,65]. A strategy proven effective in inhibiting amyloid fibril formation is to identify the self-adhesive segment of a protein (termed the “steric zipper” [66]) and deploy decoy peptides of similar sequence to mask this segment and block fibril growth. One such steric zipper was found to be residues 252–258 of DBD (LTIITLE) [28]. The 17-residue ReACp53 peptide was created by linking the related LTRITLE sequence with a poly-Arg cell-penetrating tag [67]. In primary human cancer cells harboring a variety of p53 mutations, and in organoid tumor cells bearing R248Q and R175H p53, ReACp53 reduced cytosolic p53 aggregates, facilitated p53 entry into the nucleus, and induced apoptosis in a p53 mutant-specific manner. A more recent report found similarly encouraging results in cultured prostate cancer cells and xenografts [68].

It is unclear why simply preventing p53 aggregation can apparently reactivate mutants such as R248Q and R175H in cells, given the extents to which the DNA- and zinc-binding properties of their respective DBDs are compromised. One possibility is that their inherently poor transcriptional activities can be overwhelmed by their sheer number as a large cellular pool of aggregated p53 converts to native monomers, dimers, and tetramers. Another explanation may be the abrogation of GOF. On the other side of the coin, cavity binders and zinc metallochaperones are expected to inhibit p53 aggregation by reducing the populations of partially unfolded forms that expose adhesive segments. Evidence suggests that PK083 [31] and ZMC1 [52] indeed produce this effect. In addition to these synthetic molecules, natural products including DNA [69,70], RNA [71], polyarginine [72], and the plant phenol resveratrol [73] have been shown to suppress or otherwise modulate p53 aggregation. In any case, it seems likely that the combination of aggregation inhibitors with class-specific reactivators will be an especially fruitful therapeutic strategy.

## 7. Alkylating Agents 

Some of the first small molecules that were reported to reactivate mutant p53 (e.g., CP-31398 [74] and PRIMA-1/APR-246 [75]) were initially thought to interact selectively with mutant proteins in much the same way as the compounds discussed above. It was later discovered that the active species are not the agents as synthesized, but rather reactive breakdown products that form covalent adducts with a variety of nucleophilic groups via hetero-Michael addition reactions [76]. By alkylating cellular thiol groups, the methylene quinuclidinone (MQ) decomposition product of APR-246 depletes cellular glutathione [77,78], inhibits thioredoxin and glutaredoxin systems [79,80], and inhibits ribonucleotide reductase [80], leading to oxidative stress and cancer cell death through these p53-independent pathways [81]. Nevertheless, MQ possesses p53-activating properties and APR-246 is under investigation in clinical trials enrolling cancer patients with mutant as well as unspecified p53 allele status. 

The mechanisms by which covalent modifications activate p53 are becoming clearer. A recent study of APR-246 found that MQ alkylated up to five Cys residues in WT, R273H, and R175H DBD in a concentration-dependent manner, with C277 being the most reactive [82]. Alkylation of C277 increased T_m_ of all three variants and, together with modification of C124, this global stabilization effect may account for the rescue of R175H. C277 directly interacts with DNA and is in close proximity to R273, so it is conceivable that the C277 adduct could make additional contacts with DNA to compensate for the R273H alteration [83]. The 2-sulfonylpyrimidine PK11007 [84] and HO-3867 (an analog of the plant pigment curcumin) [85] were also shown to be cysteine alkylators that preferentially attack C277 and C182 of DBD. PK11007 thermally stabilized Y220C DBD without compromising DNA binding affinity. PK11007, like PRIMA-1 [86], inhibited aggregation of mutant p53. HO-3867 refolded mutant p53 in several cancer cell lines as judged by conformation-specific antibody staining. Although both compounds possess antiproliferative activity, they demonstrated varying degrees of selectivity toward killing cells with mutant p53 (compared to null and WT p53 cell lines), consistent with their action via both p53-dependent and independent routes [84,85,87,88]. 

## 8. Challenges and Outlook 

By degrees, p53’s reputation as being undruggable is being chipped away. The affinity of Y220C-specific molecules has increased ~100 fold since PK083 and further optimizations may be anticipated. Another need is to identify new druggable pockets in other stability-class mutants. The search for DNA-contact class reactivators progressed a step with SLMP53-1. The proposed mechanism and binding pocket await validation by biochemical and structural experiments, but the discovery of a p53–DNA bridging molecule would represent a significant advancement. The clinical route for the above molecules is a classic one, taken by many drug development programs: improve the affinity and selectivity of the compounds for their targets. ZMC therapy is unique because, without a direct ZMC-p53 interaction, the pharmacological agent is not the ZMC but rather the zinc ion itself. The challenge is not to increase the strength of the ZMC-Zn^2+^ interaction—the optimal affinity range is already known and achievable [52]—but to minimize off-target effects and overcome the cellular zinc muffling response (which results in a resistance phenotype). The former problem may be addressed by ‘dialing down’ the affinity for redox-active copper and/or iron ions, and the latter by a pulsatile dosing regimen that elevates intracellular zinc levels long enough to refold reservoirs of mutant p53 but not long enough to invoke a sustained transcriptional program that elevates zinc exporter and metallothionein levels. 

In conclusion, as tremendous resources continue to be invested in drug discovery and design, it is important to recognize that only a small slice of the p53 mutational spectrum has been characterized. Missense mutations in over 190 of the 220 codons in DBD have been observed in human cancers [9]. We suggest that measuring three basic properties of these variants—*K_d_*^Zn^, *K_d_*^DNA^, and ΔG_fold_—will identify those that will respond best to class-specific molecules and help stratify patients for personalized medicine when these drugs eventually become available. Killing cancer cells by p53-directed therapy will likely entail a multi-pronged tactic that targets GOF, p53’s downstream pathways, and p53 mutant reactivation. Rescuing the DNA binding and transcriptional activities of mutant p53 is at the center of this approach and it may be considered the tip of the spear. 

## Figures and Tables

**Figure 1 biomolecules-10-00303-f001:**
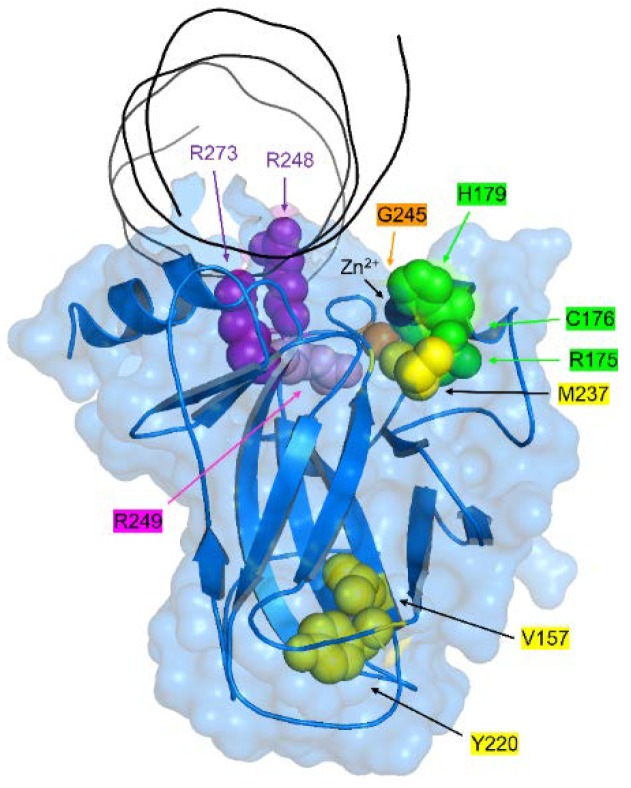
Locations of the top 10 most common somatic mutation sites in p53. The X-ray structure of wild-type (WT) DNA-binding domain (DBD) complexed to DNA is shown (PDB 1TSR). Mutational classes are colored as follows: DNA contact (purple), zinc binding (green), stability (yellow), mixed DNA-contact/stability (pink), and mixed zinc-binding/stability (orange). The zinc ion is the black sphere.

**Figure 2 biomolecules-10-00303-f002:**
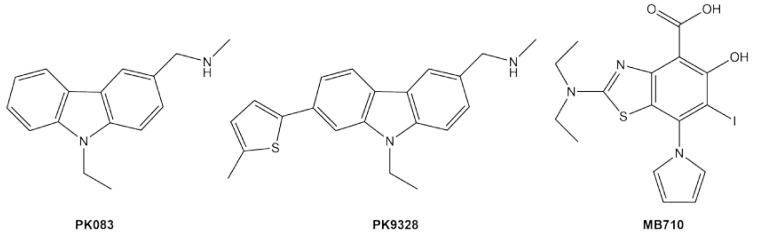
Cavity-binding compounds that target the stability class Y220C p53 mutant.

**Figure 3 biomolecules-10-00303-f003:**
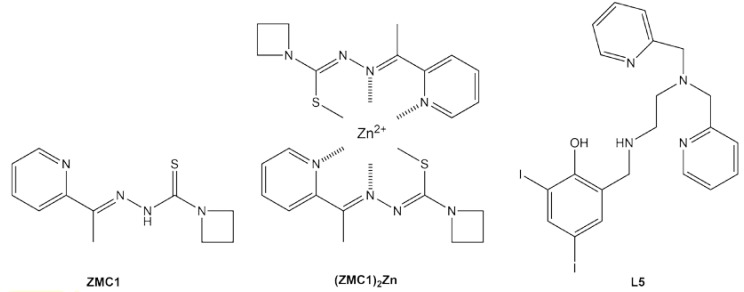
Metallochaperones that target the zinc-binding and stability class mutants of p53.

**Figure 4 biomolecules-10-00303-f004:**
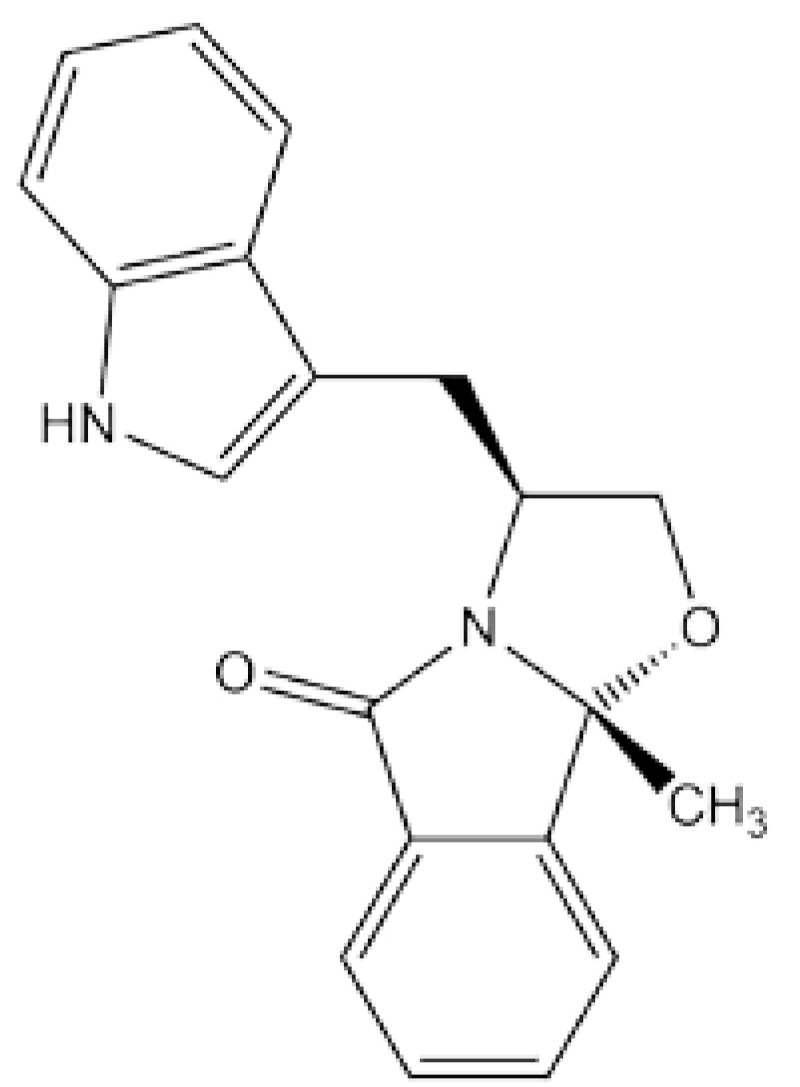
SLMP53-1 compound that targets DNA-binding class mutants of p53.

**Table 1 biomolecules-10-00303-t001:** Top10 most common somatic p53 mutation sites in IARC Database R20. If multiple mutations at a single site are among the top 50 most common, they are listed in order of decreasing occurrence.

Rank	WT	Mutants	DNA Contact	Zinc Binding	Stability	Ref.
1	R248	Gln, Trp	Yes	No	No	[20,22]
2	R273	His, Cys	Yes	No	No	[20]
3	R175	His	No	Yes	No	[23]
4	G245	Ser, Asp, Cys, Val	No	Yes	Yes	[20,23]
5	R249	Ser	Yes		Yes	[20,24]
6	Y220	Cys	No		Yes	[20]
7	C176	Phe, Tyr	No	Yes		[23]^,1^
8	H179	Arg, Tyr	No	Yes		^1^
9	V157	Phe	No		Yes	[20]
10	M237	Ile	No		Yes	[20]

^1^ Zinc-coordinating residue.

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
