# Peer review of "Follow the Mutations: Toward Class-Specific, Small-Molecule Reactivation of p53"

_biomolecules, 2020, doi:10.3390/biom10020303_

Round 1

Reviewer 1 Report

Dr. Loh writes a clear review of the ongoing efforts in the p53 re-activation field.  The major topics (DBD stabilizers, Zinc modifiers, aggregation targeting) are all covered in appropriate depth and with relevant references.  Aside from a minor quibble on the citing of the data in figure 2, I believe this review is an excellent primer for the field.

Major comments

2 introduces new data, generally this is not appropriate in a review. The author should cite this data (if published) or perhaps very similar published data.

Minor suggestions

The DBD of p53 is variously referred to as being ~220AA (abstract) or ~200AA (intro). Either is fine, but stick to one. In the introduction the author suggest that p53 is not druggable for various complex reasons and they are certainly correct. The author could also note that there are simply very few drugs that target non-enzymatic non-receptor proteins.  Developing drugs that work allosterically to activate a protein is simply hard. Lines 75/76 – please re-write this sentence to make it clear that only MDM2 (not MDMx) has functional ubiquitination activity. In lines 86-88 the author describes DNA contact mutations, a brief sentence describing the biophysical role of the loss of the positively charged arginine residues in reducing affinity for DNA might be helpful for some readers. The conclusion is concise and highlights some promising future directions but could use a little more context. What fraction of p53 mutations might be targetable?  What combination reactivators and chemotherapy might be possible/appropriate?  Are their potential resistance mechanisms to these drugs?

Author Response

I agree with all of R1's comments. Figure 2 did indeed include unpublished data and I removed it. I rewrote that discussion and included it in a new section titled "3. Crosstalk Between Mutational Classes". This helps the flow of the paper in that the mutational classes are first presented in clearer, simpler terms, which serves to preface the new section on how mutations can interact energetically. The revised MS contains no unpublished data.

The revised MS now comments on the difficulty of drugging non-receptor, non-enzymatic proteins as suggested by R1. R1's wording was concise and I took the liberty of using it in one sentence; if this is a problem please advise.

I clarified that MDM2 is the active E3 ligase that ubiquitinates p53, with MDMX providing assistance. I also mentioned that the N-terminal domain is the primary  site of MDM2 binding, as there are other binding sites outside of the NTD.

I commented on the effect of positive-to-neutral charge mutants on DNA binding as directed by R1.

The conclusion section was rewritten to address targetable mutations and combination therapies. I didn't comment on resistance mechanisms (other than the peculiar type of resistance pathway that zinc metallochaperone induces), because there is little data available and I thought it would be overly speculative.

Reviewer 2 Report

An outstanding review, and I have no hesitation with recommending publication. This is an important area and the author has provided significant detail and thoughts on moving the field forward.

Only one comment:

Figure 3: Text refers to PK083, Figure shows PK078.

Author Response

I thank R2 for his/her comments and for catching my mistake. The structure was mislabeled in the figure as PK078; it's actually PK083. I replaced it with a new figure. Note that original Fig. 2 was deleted so what was originally Fig. 3 is now Fig. 2 in the revision.

Reviewer 3 Report

A well written and informed review on p53 small molecules. I enjoyed the detailed description of p53 mutation subtypes and the compounds developed to target these. The authors clearly have an excellent grasp on the field and literature. 

A limitation of cavity binding molecules is the reduced numbers in clinical application. Perhaps this should be discussed.

Further focus could be added to the compounds that are undergoing clinical trial, i.e., APR-246 and COTI-2 as these have the greatest chance of real life impact. 

Typos: line 30, 343

Author Response

I thank R3 for his/her comments. The lack of clinical application for cavity binders is because they still have some ways to go in terms of affinity. It should hopefully be clear to the readers that 1 micromolar binding is too weak for clinical trials. With respect to COTI-2, there are little available data regarding its mechanism so there is not much more to say. The results of the single clinical trial have not yet been published. There have been numerous papers on APR246 describing its many p53-independent and p53-dependent effects on cells. However, these effects have not been sorted out and as I pointed out in the review, APR246 clinical trials are not specifically enrolling patients with mutant p53 status. I respectfully submit that given the focus of this review it is appropriate to limit the discussion to the latest findings on how APR246 modifies p53 and direct the reader to recent reviews devoted to APR246's broad anti-cancer activities, of which I cite in this paper.